# GENFUSION:
# FEED-FORWARD HUMAN PERFORMANCE CAPTURE VIA PROGRESSIVE CANONICAL SPACE UPDATES

**Youngjoong Kwon, Yao He\*, Heejung Choi\*, Chen Geng, Zhengmao Liu, Jiajun Wu, Ehsan Adeli**
Stanford University

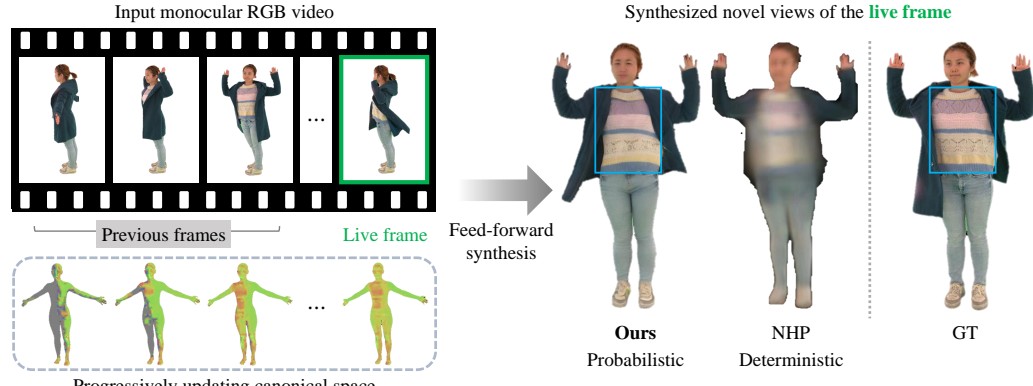

Figure 1: **GenFusion** is a feed-forward human performance capture method that progressively updates a canonical space to reconstruct humans in alignment with past observations from a monocular RGB stream. For example, given only a side-view input live frame (green box), GenFusion reconstructs the striped shirt pattern in the frontal view by *retrieving information observed in past frames from the canonical space.* Furthermore, by utilizing the probabilistic rendering, we achieve high-fidelity view synthesis. (Gray regions in the canonical space indicate unobserved areas with no information.)

## ABSTRACT

We present a feed-forward human performance capture method that renders novel views of a performer from a monocular RGB stream. A key challenge in this setting is the lack of sufficient observations, especially for unseen regions. Assuming the subject moves continuously over time, we take advantage of the fact that more body parts become observable by maintaining a canonical space that is progressively updated with each incoming frame. This canonical space accumulates appearance information over time and serves as a context bank when direct observations are missing in the current live frame. To effectively utilize this context while respecting the deformation of the live state, we formulate the rendering process as probabilistic regression. This resolves conflicts between past and current observations, producing sharper reconstructions than deterministic regression approaches. Furthermore, it enables plausible synthesis even in regions with no prior observations. Experiments on in-domain (4D-Dress) and out-of-distribution (MVHumanNet) datasets demonstrate the effectiveness of our approach.

## 1 INTRODUCTION

Imagine watching a ballet dancer performing a pirouette. At any given moment, we only see a partial view of the dancer—perhaps the side or the back. Yet, as the dancer spins gracefully, we naturally accumulate visual cues over time. By the end of the motion, our mind has pieced together a complete understanding of the dancer's appearance, despite never having seen every part at once.

---

\*Equal contribution

Inspired by this intuition, we propose a feed-forward human performance capture method that renders high-fidelity novel views of a performer from a monocular RGB stream. At the core of our approach is a progressively updated canonical space that integrates new observations over time. To achieve this, we leverage 4D correspondences defined by a human template model (e.g., SMPL-X (Pavlakos et al., 2019)) to establish consistent mappings between the live frames and the canonical space. This use of SMPL-X is not our core contribution, but rather a means to enable temporal alignment. Specifically, as each new frame arrives, we extract live-frame features and aggregate them into the canonical feature map. Over time, the canonical space is gradually completed and serves as a temporal context bank, enabling reconstructions that reflect previously observed appearances, even when they are not directly visible in the current frame. This stands in stark contrast to per-frame reconstruction methods, which either regress toward averaged appearances seen in the training distribution (Saito et al., 2019; Choi et al., 2022; Zhao et al., 2022; Hu et al., 2023; Kwon et al., 2021; 2025; Li et al., 2024a) or generate realistic but unrelated details to prior observations (Zhu et al., 2024b; Hu, 2024; Madges et al., 2017; Li et al., 2024b; Ho et al., 2024; Zhang et al., 2024; He et al., 2024; Shao et al., 2024a; Xu et al., 2024; Qiu et al., 2025b;a).

To synthesize novel views that are both grounded in the canonical context and coherent with the live deformation state (*i.e.*, deformation state of the current input frame), an intuitive approach is to apply deterministic regression with pixel-wise supervision (e.g., mean squared error between the predicted novel view and ground truth). However, this formulation struggles when past and current frames exhibit misalignments. For example, a shirt seen in a leaning pose in the past may not perfectly align with the upright pose in the live frame, causing the model to suppress high-frequency details and produce blurry outputs (see Fig. 1-NHP and Fig. 3).

To address this, we formulate our rendering process as probabilistic regression using a diffusion-based model (Rombach et al., 2022). While the usage of probabilistic rendering alone is not our primary contribution, it plays a key role in enabling effective use of the canonical context. Rather than enforcing exact pixel-wise alignment, the probabilistic regression supervision encourages perceptually realistic synthesis. This allows the model to incorporate semantically relevant cues from the canonical space, such as textures or patterns, even when misalignments in pose or geometry exist. Moreover, the probabilistic nature of our model enables plausible synthesis in regions where the canonical space lacks prior observations—for example, rendering a frontal view even when no frontal information has been previously stored (see Fig. 2 canonical space visualization).

We validate the effectiveness of our approach on both in-domain (Wang et al., 2024, 4D-Dress) and out-of-distribution (Xiong et al., 2024, MVHumanNet) datasets. Comparisons with both deterministic (per-frame and temporal) and probabilistic regression baselines demonstrate that our method benefits from the combination of progressively evolving canonical space and the probabilistic rendering.

## 2 RELATED WORK

Deep learning-based methods have enabled human performance capture from sparse or single views, addressing some of these accessibility limitations of traditional methods. Certain optimization methods (Habermann et al., 2019; 2020; 2021; Zhu et al., 2024a; Pang et al., 2024; Xiang et al., 2023; Shetty et al., 2024) model high-quality non-rigid deformations corresponding to pose changes; however, they are heavily dependent on the quality of pre-captured template mesh. Recent optimization-based methods (Peng et al., 2023; 2024; Geng et al., 2023; Weng et al., 2022; Hu et al., 2024b; Qian et al., 2024; Ma et al., 2024; Moon et al., 2024; Hu et al., 2024a; Shao et al., 2024b; Li et al., 2024c) instead leverage human templates (Pavlakos et al., 2019) to optimize vertex point features, which are then rendered into novel views and poses. Yet, they cannot generalize to new subjects.

More recent feed-forward methods (Huang et al., 2020; Saito et al., 2019; 2020; He et al., 2020; Li et al., 2020; He et al., 2021) bypass the need for test-time optimization by conditioning on the image features or pixel-aligned features. However, these methods struggle with complex poses due to limited understanding of 3D structure. To address complex human reconstruction in sparse input settings, a line of methods (Zhao et al., 2022; Chen et al., 2022; Choi et al., 2022; Hu et al., 2023; Pan et al., 2024; Li et al., 2024a) incorporates 3D human templates to aggregate features across the view axis. However, these approaches struggle to extrapolate unobserved details, particularly in single-view scenarios. NHP (Kwon et al., 2021) uses a human template to aggregate temporal information and compensate for insufficient observations. However, it produces blurry, averaged

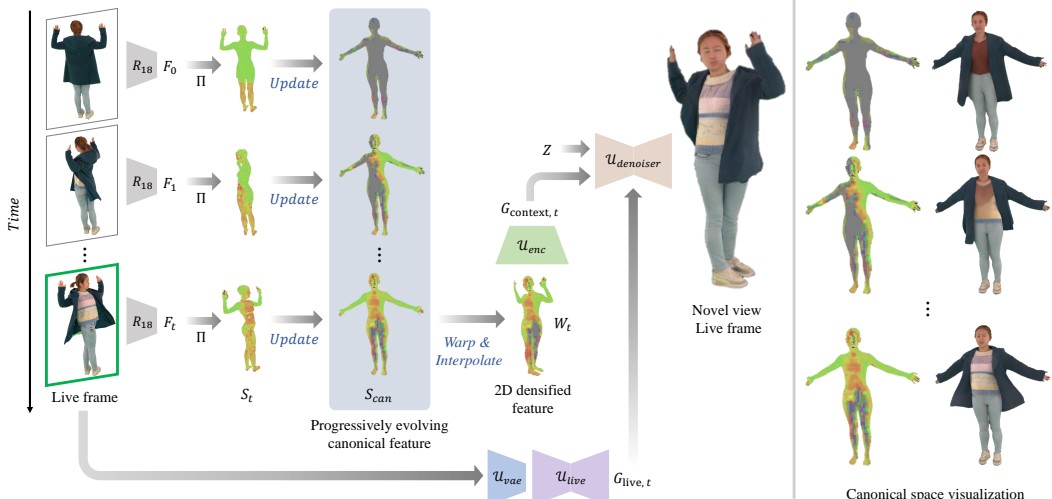

Figure 2: **GenFusion** renders novel views of live frames in a feed-forward manner from a monocular RGB stream. Given a live frame $I_t$, feature map $F_t$ is extracted and aligned to the SMPL-X template mesh, resulting in vertex-aligned feature set $S_t$. $S_t$ is fused into the canonical feature set $S_{\text{can}}$, updating temporal history. The canonical feature set $S_{\text{can}}$ is then warped and densified into the live space, forming the input $G_{\text{context,t}}$ for the denoising network $\mathcal{U}_{\text{denoiser}}$. The denoising network denoises the noisy image $Z$ into the final novel view live frame, conditioned on the live frame's state $G_{\text{live,t}}$. The right side shows the progressive update of the canonical space. The first column shows the feature map; the second shows its RGB rendering. Our method synthesizes realistic details even without observations (top row) and refines the canonical space as more frames are incorporated.

appearances to avoid misalignment penalties between past and current frames, which is a known challenge for deterministic regression models.

The emergence of probabilistic regression models (*e.g.*, diffusion model (Rombach et al., 2022)) has provided new opportunities for realistic single-view synthesis by harnessing their generative capabilities to hallucinate unobserved details (Liu et al., 2023; 2024; Long et al., 2024). While these methods can produce visually compelling results, they often introduce distortions and artifacts. Some approaches (Huang et al., 2023; 2024; Zhang et al., 2024; Ho et al., 2024; Hu, 2024; Madges et al., 2017; Xu et al., 2024; He et al., 2024; Shao et al., 2024a; Li et al., 2024b; Zhu et al., 2024b; Qiu et al., 2025a;b; Sengupta et al., 2024) integrate human templates to mitigate these distortions; however, as per-frame methods that do not incorporate temporal history, they often generate details that are not aligned with past observations.

In this work, we propose to leverage temporal information from incoming frames to progressively build a comprehensive canonical space. This canonical space provides context for synthesis, even when certain regions are not visible in the current frame, enabling synthesis that aligns with previous frames. Furthermore, to be able to effectively utilize the context stored in the canonical space, we employ the probabilistic rendering model.

## 3 METHOD

Given a monocular RGB video of a performer, corresponding fitted human templates (*i.e.*, SMPL-X (Pavlakos et al., 2019)), and camera parameters for both input and novel views, our goal is to generate novel views of the live frame (*i.e.*, input frame at the current time step) in a manner that *aligns with past observations*. To achieve this, we introduce a feed-forward method that progressively updates the shared canonical space and utilizes it as a context for synthesizing live novel views. Figure 2 illustrates how our framework processes input live frames by aligning them to the canonical space and updating the canonical space with a visibility-based fusion process (Section 3.2). The updated canonical space is then warped into the live space and rendered into novel view frames conditioned on the live deformation (Section 3.3). Section 3.4 provides training and inference details.

### 3.1 MOTIVATION

Capturing human performance from a single monocular RGB stream poses challenges due to insufficient observations. While each frame provides only a partial observation, different frames over time can reveal additional details as the subject moves in front of the camera. By continuously updating the canonical space with each new frame, we can build a progressively complete representation, compensating for missing information in individual live frames. An additional challenge involves transforming the canonical space back into the live space. Traditional methods rely on optimizing an SE(3) transformation to warp the canonical space to the live space, which is difficult with only monocular input and dynamic subjects undergoing non-rigid deformations. This work, therefore, emphasizes gradually completing the canonical space and effectively renders the live space out of it.

### 3.2 CANONICAL SPACE CONSTRUCTION AND PROGRESSIVE UPDATING

**Initialization.** To begin, the canonical space feature set is initialized as $S_{\text{can}} = \mathbf{0} \in \mathbb{R}^{M \times L}$, where $M$ denotes the number of vertices in the human template mesh (*i.e.*, SMPL-X) and $L$ is the channel dimension (e.g., $L$=256). Additionally, we employ a visibility frequency map $V_{\text{can}}$, initialized as $\mathbf{0} \in \mathbb{R}^{M \times 1}$, to record the visibility frequency of each vertex over time.

**Aligning Live Frames to the Canonical Space.** To integrate live frame information into the canonical space, we align the current live frame at time $t$ to the canonical space using a fitted human template model. This alignment also incorporates the temporal history of previously integrated frames and their features, allowing the canonical representation to be progressively updated through feature accumulation and fusion over time. The use of the template model establishes 4D correspondences across frames, enabling effective alignment between the live frames and the canonical space.

For the current live frame $I_t$, we extract hierarchical feature maps $F_t$ from the first three layers of ResNet-18. These feature maps progressively downsample the resolution to 1/2, 1/4, and 1/8 of the input image. This multi-level representation captures both fine-grained details (e.g., textures) and high-level semantic context (e.g., region-level features). Notably, the receptive fields in ResNet allow these feature maps to encode contextual information such as clothing or hair that may extend beyond the SMPL-X surface. The extracted feature map $F_t$ is then aligned to the SMPL-X template mesh. This process begins with a 3D-to-2D projection operation $Proj$ which maps the 3D world coordinates of the SMPL-X vertices $X_t$ onto the 2D image plane using the input view camera parameters $C_{\text{input}}$. Next, a bilinear sampling operation $\Pi$ samples the corresponding features from $F_t$ at the 2D projected vertex locations. This results in a vertex-aligned feature set $S_t \in \mathbb{R}^{M \times L}$, where $M$ is the number of SMPL-X vertices and $L$ is the feature dimension. Formally, $S_t = \Pi(F_t, Proj(X_t, C_{\text{input}}))$.

The canonical feature set $S_{\text{can}} \in \mathbb{R}^{M \times L}$ is dynamically updated by incorporating the features $S_t$ from each live frame, progressively refining the representation. To achieve this, we utilize a visibility frequency map $V_{\text{can}} \in \mathbb{R}^{M \times 1}$, which accumulates the per-vertex visibility over time. This map ensures that the contributions of both historical and current features are appropriately weighted based on their observed frequency. For the current live frame at time $t$, the visibility map $V_t \in \mathbb{R}^{M \times 1}$ is computed, and the corresponding canonical features $S_t \in \mathbb{R}^{M \times L}$ is incorporated into $S_{\text{can}}$ as follows:

$$S_{\text{can}} = \frac{(S_t \cdot V_t) + (S_{\text{can}} \cdot V_{\text{can}})}{\max(V_t + V_{\text{can}}, 1)}. \tag{1}$$

After updating the canonical feature, the visibility frequency map is updated to include the current frame's visibility $V_{\text{can}} = V_{\text{can}} + V_t$. This approach dynamically weighs the contribution of each vertex in $S_t$, ensuring a balanced and progressively refined canonical space that integrates both historical context and current observations. The progressively updating canonical space serves as a context bank, enabling novel views of the live frame to be rendered in alignment with past observations, even when those regions are not directly visible in the current live frame.

### 3.3 PROBABILISTIC REGRESSION OF THE LIVE SPACE

The goal of this step is to reconstruct the live frame by synthesizing realistic details that are aligned with the context stored in the canonical space. This task is particularly challenging with a monocular input stream, especially when dealing with dynamic, non-rigid deformations such as loose garments.

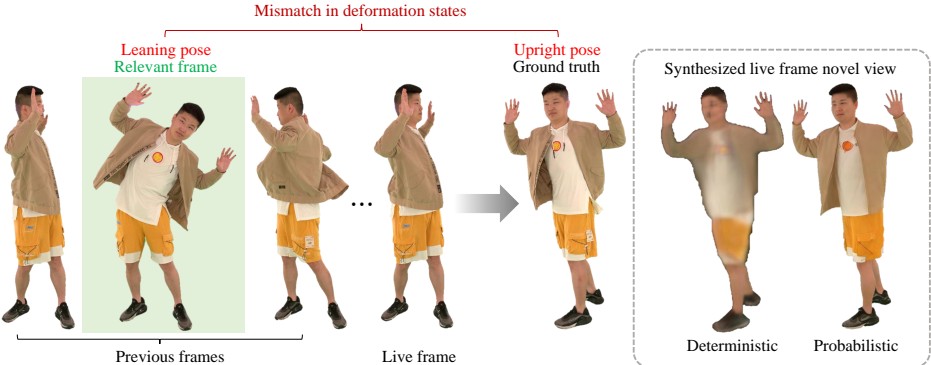

Figure 3: Deterministic regression supervision (e.g., pixel-wise loss) penalizes deformation mismatch, leading to blurry outputs, while probabilistic regression supervision focuses on perceptually realistic synthesis rather than pixel-wise mismatch.

**Limitations of Deterministic Regression.** Deterministic regression-based approaches, which directly map canonical features to the live frame using pixel-wise reconstruction losses like $\ell_1$ or MSE, often struggle to handle high-frequency details. These losses penalize pixel-level differences between the predicted frame and the ground truth, making them particularly problematic when observed details change dynamically over time (see Figure 3). For example, consider a scenario where the appearance of certain high-frequency details, such as wrinkles or fabric deformations, evolves continuously due to non-rigid motion. If the available input only provides partial or incomplete views of the subject over time, the model must rely on prior observations to infer unobserved regions. However, when the ground truth for these regions reflects updated details that differ from the prior state, pixel-wise losses penalize the model for its reliance on previous observations. This discourages the model from predicting fine-grained details that are prone to temporal variation. Instead, it learns to predict low-frequency information, such as general colors or shapes, resulting in pixel-averaged, blurry outputs. To address these limitations, we adopt a probabilistic regression model, *i.e.*, diffusion model (Rombach et al., 2022). Rather than focusing on architectural novelty in probabilistic rendering, our contribution lies in demonstrating *how a carefully designed canonical context can significantly improve synthesis quality when paired with off-the-shelf diffusion models*. Therefore, we use a standard pre-trained VAE model and Stable Diffusion model provided by Hugging Face.

**Warping and Densifying Canonical Features into Live Space.** To adapt the canonical feature set $S_{\text{can}}$ into the live space's pose, we warp it using the 3D world coordinates of the SMPL-X mesh vertices $X_t$. This warping step transforms the canonical features into the live pose, represented as $Warp(S_{\text{can}}, X_t)$. For live frame reconstruction, we project this vertex-aligned representation into the novel view, conditioned on the camera parameters $C_{novel}$. While $S_{\text{can}}$ is warped to the live pose, it remains a sparse, vertex-aligned representation ($S_{\text{can}} \in \mathbb{R}^{M \times L}$, where $M$ is the number of SMPL-X vertices). To enable dense frame reconstruction, we perform barycentric interpolation which renders the sparse features into a dense 2D image: $W_t$ as $W_t = Interpolate(Warp(S_{\text{can}}, X_t), C_{novel})$. Here, $Interpolate$ densifies the sparse vertex features into a dense 2D feature image aligned with the novel view. The resulting $W_t$ incorporates rich temporal context aggregated from $S_{can}$. This dense representation forms the foundation for reconstructing the live frame.

**Live Space Reconstruction.** To reconstruct the live frame, we adopt a probabilistic regression process that integrates warped canonical features $W_t$ and the deformation state of the reference live frame $I_t$. This process involves encoding the dense canonical features, capturing the live frame's deformation state, and synthesizing the final output through a diffusion-based denoising.

The dense canonical feature $W_t$, obtained through barycentric interpolation of the warped canonical features, is first encoded into a compact representation $G_{\text{context},t}$ using $\mathcal{U}_{\text{enc}}$, a network composed of convolutional and self-attention layers: $G_{\text{context},t} = \mathcal{U}_{\text{enc}}(W_t)$. This encoding enriches the canonical features, providing contextual information necessary for synthesis. Simultaneously, the deformation state of the reference live frame $I_t$ is captured to represent the subject's dynamic, non-rigid changes. This is achieved by first encoding $I_t$ using a variational autoencoder $\mathcal{U}_{\text{vae}}$, followed by a network $\mathcal{U}_{\text{live}}$ which refines the encoded features into a live state-aware representation: $G_{\text{live},t} = \mathcal{U}_{\text{live}}(\mathcal{U}_{\text{vae}}(I_t))$. To synthesize the final live frame, we employ a denoising network $\mathcal{U}_{\text{denoiser}}$, inspired by Stable Diffusion

(Rombach et al., 2022). This network operates on the encoded canonical features $G_{\text{context},t}$, the live state-aware features $G_{\text{live},t}$, and a noisy latent input $Z_t$, progressively refining the input to reconstruct the live frame: $\mathcal{U}_{\text{denoiser}}(Z_t, G_{\text{context},t}, G_{\text{deform},t})$.

Our probabilistic framework is trained using a diffusion-based approach. At each diffusion step $t$, the denoiser predicts the noise $\epsilon$ added to the ground-truth latent vector $Z$. The training objective minimizes the difference between the predicted and the ground-truth noise: $\mathcal{L} = \mathbb{E}\left[\|\epsilon - \mathcal{U}_{\text{denoiser}}(Z_t, G_{\text{context},t}, G_{\text{live},t}, i)\|^2\right]$, where $Z_t = \alpha_t Z + \sigma_t \epsilon$. Here, $Z$ is the ground-truth latent, $\epsilon \sim \mathcal{N}(0, I)$ represents Gaussian noise, and $\alpha_i$ and $\sigma_i$ are diffusion parameters that define the noise level at diffusion timestep $i$. This objective ensures that the model learns to synthesize realistic frames by leveraging past observations to resolve ambiguities inherent in monocular inputs.

### 3.4 TRAINING AND INFERENCE

**Training.** The training input consists of a reference live frame at time $t$ from the input view camera $C_{\text{input}}$ and $N$ preceding frames sampled at a temporal stride $K$. Formally, the input sequence is $\{I_t, I_{t-K}, I_{t-2K}, \ldots, I_{t-NK}\}$, where $N$ is the number of preceding frames and $K$ is the temporal stride. During training, $N$ is set to 10, and $K$ is randomly sampled from the set $\{1, 5, 10\}$, introducing variability in temporal spacing to encourage the canonical features $S_{\text{can}}$ to capture a rich temporal context. A stride of $K = 1$ corresponds to consecutive frames, matching the inference setting, while $K = 5$ or 10 expands the temporal spacing to include a broader context for $S_{\text{can}}$. This strategy ensures the model learns to rely on $S_{\text{can}}$ as a robust context for live frame reconstruction. The model is trained for 100,000 iterations with a learning rate of $1 \times 10^{-5}$, using a batch size of 1 on a single NVIDIA L40S GPU. The total training time is approximately 24 hours. Certain components such as the VAE and ResNet-18 feature extractor are frozen, while the feature image encoder $\mathcal{U}_{\text{enc}}$, live frame encoder $\mathcal{U}_{live}$, and denoising network $\mathcal{U}_{denoiser}$ remain trainable. The output of the training process is a novel view synthesis of the reference live frame $I_t$ from a target camera view $C_{\text{novel}}$. The training objective is an MSE loss between the ground-truth noise $\epsilon$ and the predicted noise at each denoising timestep $i$.

**Inference.** During inference, the model processes input frames sequentially to reconstruct live frames. For each frame $I_t$ at the current time $t$, consecutive frames ($K = 1$) are used, reflecting the sequential nature of live streaming scenarios. The canonical feature $S_{\text{can}}$ is dynamically updated with each new frame. Our setup uses $T = 10$ diffusion steps.

## 4 EXPERIMENTS

### 4.1 BASELINES, DATASETS, AND METRICS

**Baselines.** We compare ours against the following feed-forward human reconstruction baselines: (1) per-frame deterministic regression methods, SHERF (Hu et al., 2023) and GHG (Kwon et al., 2025); (2) a temporal deterministic regression method, NHP Kwon et al. (2021); and (3) per-frame probabilistic methods, Champ (Zhu et al., 2024b), AniGS Qiu et al. (2025b), LHM Qiu et al. (2025a), and SiFU (Zhang et al., 2024). To highlight the efficiency of our feed-forward design, we additionally compare with GauHuman (Hu et al., 2024b), which performs subject-specific optimization.

**Datasets.** We use the THuman2.1 (Yu et al., 2021) and 4D-Dress (Wang et al., 2024) datasets for training, with all comparison methods trained on the same dataset and protocol. We reserve 30 subjects of the 4D-Dress dataset for testing to assess in-domain generalizability. For cross-dataset evaluation, we test on 30 representative MVHumanNet sequences. For evaluating in-the-wild generalizability, we use the TikTok (Jafarian & Park, 2021) dataset, where foreground masks are obtained with RemBG (Gatis, 2022), and SMPL-X fits with SMPLest-X (Yin et al., 2025).

**Metrics.** To evaluate frame-level quality, we use PSNR. To capture human perception, we also employ perceptual metrics: LPIPS-VGG (Zhang et al., 2018). To assess how well the synthesis aligns with past observations, we use Fréchet Video Distance (FVD) (Unterthiner et al., 2018), computed on sequentially synthesized frames from a single viewpoint and averaged over the novel views.

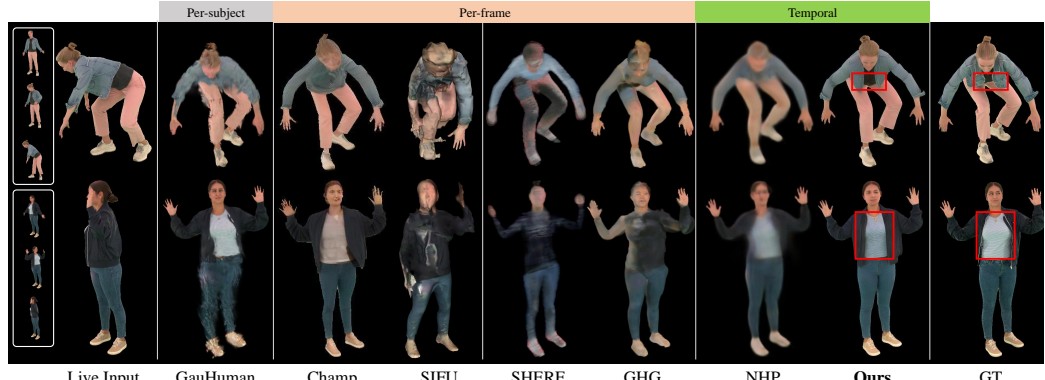

Figure 4: **In-domain generalization results on the 4D-Dress dataset.** Monocular input stream is shown in the white boxes. Our method effectively reconstructs novel views that align with past observations. Per-frame probabilistic methods (Champ, SIFU) fail to reconstruct the details observed in prior observations. Per-frame deterministic regression methods (SHERF, GHG) struggle with synthesizing unobserved details. Temporal deterministic method NHP can leverage temporal context but produces blurry outputs. GauHuman requires per-subject optimization but delivers lower visual quality and lacks generalizability.

Table 1: **In-domain generalization results on the 4D-Dress dataset.** Our method demonstrates superior performance on perceptual metrics, highlighting the effectiveness of leveraging temporal context and probabilistic rendering. For readability, LPIPS-VGG is scaled by a factor of 1000.

| Method | Generalizable | Temporal Context | Synthesis Objective | PSNR ($\uparrow$) | LPIPS-VGG ($\downarrow$) | FVD ($\downarrow$) |
|---|---|---|---|---|---|---|
| *GauHuman Hu et al. (2024b)* | ✗ (Per-subject) | ✗ | Deterministic | 23.19 | 83.34 | 500.8 |
| *Champ (Zhu et al., 2024b)* | ✓ | ✗ | Probabilistic | 19.37 | 98.61 | 254.5 |
| *SHERF (Hu et al., 2023)* | ✓ | ✗ | Deterministic | 21.86 | 86.34 | 735.3 |
| *GHG (Kwon et al., 2025)* | ✓ | ✗ | Deterministic | 24.50 | 75.60 | 502.93 |
| *NHP (Kwon et al., 2021)* | ✓ | ✗ | Deterministic | 24.72 | 96.26 | 630.0 |
| **Ours** | ✓ | ✓ | Probabilistic | **25.07** | **62.97** | **176.7** |

## 4.2 IN-DOMAIN GENERALIZATION

In Fig. 4 and Tab. 1, we evaluate in-domain generalization on the 4D-Dress dataset, comparing against feed-forward methods that are per-frame probabilistic (Champ, SIFU), deterministic (per-frame SHERF, GHG; temporal NHP), and one per-subject optimization method (GauHuman). All methods except GauHuman are trained on the THuman 2.1 and 4D-Dress training splits and evaluated on the unseen 4D-Dress test split. GauHuman is optimized for each test subject. For SIFU, which focuses on 3D mesh reconstruction rather than rendering, we report only qualitative results.

Champ, a frame-based probabilistic method, produces sharp and visually appealing results at the frame level due to its generative capabilities. However, its lack of temporal context leads to the generation of details that are irrelevant to past observations. For instance, as shown in the top row of Fig. 4, Champ neglects the black shirt underneath the blue jacket. Similarly, in the bottom row, Champ fails to render the blue shirt, as it is occluded in the current view, and Champ does not incorporate prior observations. SIFU also generates results that do not match past observations.

SHERF and GHG, frame-based deterministic regression methods, struggle with monocular input due to their inability to hallucinate unobserved details, resulting in averaged outputs (Fig. 4). While they achieve relatively high PSNR scores (Tab. 1), likely due to pixel-level supervision, their perceptual scores are poor. NHP, a temporal deterministic method, leverages past frames but still produces blurry results, as it suppresses sharp features to avoid misalignment penalties.

GauHuman uses per-subject optimization, tailoring models to individual test subjects. Despite its subject-specific focus, its visual quality is not on par with other methods. Also, it lacks generalizability and requires optimization for each new subject.

As shown in Tab. 1, our method outperforms all baselines on perceptual metrics, demonstrating its ability to render high-quality frames that align with past observations. These results underscore the effectiveness of integrating temporal context and probabilistic rendering for robust human performance capture in monocular settings.

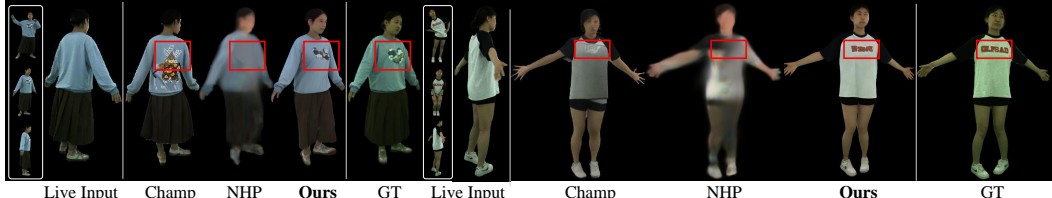

| Live Input | Champ | NHP | **Ours** | GT | Live Input | Champ | NHP | **Ours** | GT |

Figure 5: **Cross-dataset generalization results on the MVHumanNet dataset.** Left column presents the monocular input stream. Champ hallucinates irrelevant details when those regions are invisible in the current live input frame. In contrast, our method generates details relevant to past observations.

Table 2: **Cross-dataset generalization on MVHumanNet.** Our method effectively leverages temporal context to synthesize details coherent with frame history. Champ, while visually appealing, generates details that are inconsistent with past observations, as reflected in the FVD metric.

| Method | Generalizable | Temporal Context | Synthesis Objective | PSNR (↑) | LPIPS-VGG (↓) | FVD (↓) |
|---|---|---|---|---|---|---|
| *Champ (Zhu et al., 2024b)* | ✓ | ✗ | Probabilistic | 21.06 | 97.61 | 674.1 |
| *NHP (Kwon et al., 2021)* | ✓ | ✓ | Deterministic | **22.25** | 131.91 | 1321.4 |
| **Ours** | ✓ | ✓ | Probabilistic | 21.25 | **87.85** | **436.9** |

## 4.3 CROSS-DATASET GENERALIZATION

To evaluate the cross-dataset generalizability, we test on the MVHumanNet dataset (Xiong et al., 2024). We select 30 videos with notable subject motion and clothing variations, such as distinct patterns or differences between frontal and back views. These criteria ensure a diverse temporal context, providing a meaningful evaluation of our method's ability to leverage temporal information. We compare our method with the most relevant baseline NHP and Champ.

As shown in Fig. 5 and Tab. 2, our method exhibits strong generalization, producing details that align with past observations. In contrast, NHP, while able to consult the past history, generates blurry results. On the other hand, Champ, while having sharp results at the frame level, generates details inconsistent with previous frames, similar to its performance in the in-domain setting.

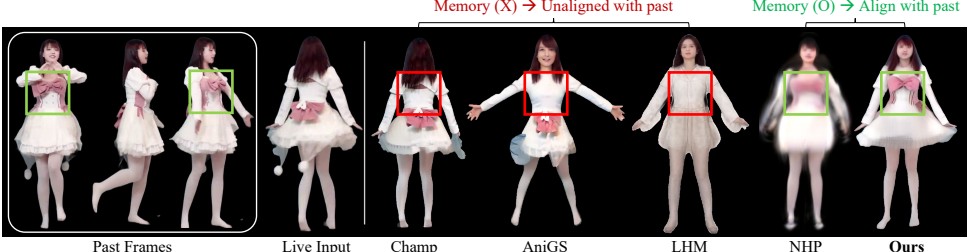

| Past Frames | Live Input | Champ | AniGS | LHM | NHP | **Ours** |

Figure 6: **In-the-wild generalization on TikTok.** Task: generate a frontal view from a back view. Champ copies the input, NHP is blurry, and ours produces a clear, observation-faithful frontal view.

## 4.4 IN-THE-WILD GENERALIZATION

We evaluate in-the-wild generalizability on the TikTok dataset (Jafarian & Park, 2021). Since GT is not available, we report only qualitative results in Fig. 6. Although the frontal view is not available in the current frame, our method reconstructs details consistent with past observations (e.g., the pink ribbon) using canonical space memory. In contrast, per-frame probabilistic methods (Champ, AniGS, LHM) generate details that are inconsistent with past observations.

## 4.5 ABLATION

Fig. 7 and Tab. 3 shows ablation studies on the 4D Dress dataset. The experimental setup follows the same training and testing protocol as in Sec. 4.2. We compare four variants of our method: (a) *No Temporal Context* replaces the temporally aggregated feature $W_t$, derived from canonical space feature $S_{can}$, with a current live frame normal map, ignoring any temporal context or history; (b) *No Feature Context* replaces the encoded pixel-aligned features ($F_t$) used to construct $S_{can}$ with raw RGB pixel

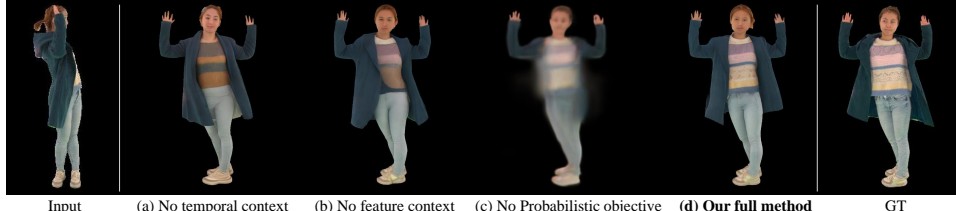

| Input | (a) No temporal context | (b) No feature context | (c) No Probabilistic objective | **(d) Our full method** | GT |

Figure 7: **Ablation results on the 4D-Dress dataset**. (a) Removing temporal context leads to inconsistencies, while (b) using raw RGB values instead of encoded features limits detail recovery. (c) A deterministic regression struggles to capture fine-grained details and dynamic deformations. (d) Our full method, combining all components, achieves superior performance in terms of visual quality and coherence to temporal context.

Table 3: **Ablation studies on the 4D-Dress dataset**. (a) Without temporal context, frame-level quality is maintained but temporal coherence degrades. (b) Using raw RGB values as input leads to a decline in perceptual quality due to the loss of fine details. (c) The deterministic regression objective exhibits limitations in capturing high-frequency details. (d) Our full method effectively leverages rich temporal observations encoded in the feature representation and the probabilistic regression to achieve high-quality results align with past observations.

| Method | Temporal Context | Synthesis Objective | PSNR ($\uparrow$) | LPIPS-VGG ($\downarrow$) | FVD ($\downarrow$) |
|---|---|---|---|---|---|
| *(a) No temporal context* | None (Normal map of current frame) | Probabilistic | 25.03 | 63.34 | 177.4 |
| *(b) No feature context* | Raw RGB values | Probabilistic | 24.37 | 64.51 | 191.9 |
| *(c) No probabilistic objective* | Encoded features ($S_{can}$) | Deterministic (Pixel MSE) | **25.23** | 95.70 | 572.3 |
| *(d) Our full method* | Encoded features ($S_{can}$) | Probabilistic | 25.07 | **62.97** | **176.7** |

values from the input frame. The temporal aggregation process follows the same visibility-weighted algorithm described in Eq. (1); (c) *No Probabilistic Objective* retains the temporal context but replaces the probabilistic diffusion-based objective with a pixel-wise deterministic regression loss; (d) *Full Method* incorporates both temporal aggregation using encoded features and the diffusion-based probabilistic objective, forming the complete version of our proposed approach.

**Effect of Temporal Context.** Fig. 7(a) and Tab. 3(a) show results without temporal context. While frame-level quality remains comparable, the model produces details inconsistent with prior observations. For example, it infers the stripe pattern from the visible region but generates mismatched patterns in occluded areas. This highlights the importance of temporal context in reconstructing unobserved details from monocular input.

**Effect of Feature Context.** Using raw RGB values instead of features (Fig. 7(b), Tab. 3(b)) allows the method to capture some temporal information. However, raw RGB values lack the spatial richness of encoded features. Encoded features enable our full method to recover details better aligned with past observations, particularly in occluded regions as in Fig. 7(d).

**Effect of the Probabilistic Regression Objective.** Using a deterministic loss (Fig. 7(c), Tab. 3(c)) maintains temporal context but struggles with high-frequency details, as discussed in Sec. 3.3. While the method renders regions like the striped sweater, it produces blurry outputs due to the limitations of pixel-wise losses, which discourage fine-grained details that frequently vary over time.

**Full Method.** Our full method (Fig. 7(d), Tab. 3(d)) outperforms all ablated variants, highlighting the combined benefits of temporal context, encoded features, and the diffusion-based objective.

## 5 CONCLUSION

We present a feed-forward human performance capture method from a monocular RGB stream. By updating live observations into a shared canonical space, our method compensates for the limitations of monocular input, building a complete representation over time. Further, by leveraging a probabilistic regression-based training objective, our method enables high-fidelity live space renderings that capture sharp details and handle non-rigid deformations effectively. Evaluations on diverse datasets confirm that our method achieves superior rendering quality that align with past observations compared to per-frame or deterministic methods.

ACKNOWLEDGMENTS

This work is in part supported by the Stanford Institute for Human-Centered AI (HAI), NIH Grant R01-AG089169, NSF RI #2211258, ONR MURI N00014-22-1-2740, and Panasonic.

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
