# OpenReview forum: "GenFusion: Feed-forward Human Performance Capture via Progressive Canonical Space Updates"
_ICLR.cc/2026/Conference — ICLR 2026 Poster_

### Official Review · Reviewer_BCER · 2025-10-29

**Soundness:** 3
**Presentation:** 3
**Contribution:** 3
**Rating:** 6
**Confidence:** 4

**Summary:**

This paper presents a feed-forward human performance capture method that reconstructs and synthesizes novel views of human subjects from a monocular RGB video stream. The key idea is to maintain a progressively updated canonical space, where visual features from consecutive frames are accumulated and fused using visibility-based weighting. This canonical space serves as a temporal memory that compensates for missing observations in the current live frame. To render novel views consistent with both past observations and current deformation, the authors employ a probabilistic regression formulation based on diffusion models. The probabilistic rendering alleviates blurriness caused by misalignment between frames and enables plausible synthesis even for previously unobserved regions.
The method is evaluated on several datasets and compared against deterministic and probabilistic baselines such as NHP, SHERF, Champ, and GauHuman, showing improved perceptual quality and temporal consistency.

**Strengths:**

The paper elegantly combines canonical-space-based temporal accumulation with a diffusion-based rendering framework. This hybrid approach effectively mitigates the limitations of deterministic regression. Unlike other optimization-based methods , this approach runs efficiently and generalizes to unseen subjects without per-frame or per-scene training.


The probabilistic rendering formulation allows plausible completion of unseen regions and reduces the dependency on perfectly aligned geometry

**Weaknesses:**

The method relies on accurate SMPL-X alignment to build temporal correspondences, but the paper does not quantify how fitting errors or template inaccuracies affect the reconstruction quality. Including such results will make this paper stronger.

Although the probabilistic rendering can hallucinate plausible details, the canonical space itself remains template-driven, which may limit fidelity for highly non-rigid clothing dynamics, which can be seen from the demo video.

**Questions:**

What is the inference time of the proposed method?

How long an sequence can it support?

---

> ### Author Response · Authors · 2025-11-21
>
> We thank the reviewer for the constructive review.
>
> **Also, please refer to the updated supplementary material titled “tiktok comparison video.mp4”.**
>
> ---
>
> **Q1. How does inaccurate SMPL X fitting affect the reconstruction quality?**
>
> To evaluate the impact of such errors, we conducted an experiment using estimated SMPL-X meshes from an off-the-shelf monocular method (SMPLest X [1]) instead of GT fits. Compared to our original setup, we observed an overall performance drop of 7.1% in PSNR, 1.4% in LPIPS VGG, and 2.3% in FVD. The larger drop in PSNR reflects pixel space misalignments caused by depth ambiguity in monocular pose estimation, while the relatively stable perceptual metrics such as LPIPS and FVD suggest that our method remains robust to moderate SMPL-X estimation noise.
>
> Also, as can be seen in the main paper Fig. 6 and in the **updated supplementary video on the in-the-wild TikTok dataset (tiktok comparison video.mp4)** where we use the off-the-shelf monocular SMPL-X estimator SMPLest-X, the result shows degradation compared to our original supplementary video results on 4D Dress and MVHumanNet.
>
> We will include this discussion in the revision.
>
> [1] Yin and Cai et al., "SMPLest-X: Ultimate Scaling for Expressive Human Pose and Shape Estimation", arXiv 2025
>
> ---
>
> **Q2. Template based mapping limits fidelity for highly non rigid clothing dynamics.**
>
> It is true that template driven canonical-live space mapping is limited, especially for highly non-rigid clothing.
>
> Initially, we experimented with using dense tracking such as SpatialTracker [2] instead of SMPL-X mapping. For example, we attempted to track inner clothing pixels across frames so that when they become occluded by other semantics such as a jacket, we could correctly denote their visibility as invisible. However, dense tracking required significant processing time and memory and could not support resolutions of 512x512 or higher. In addition, dense tracking of highly non-rigid dynamic motions such as swinging jackets or skirts was even more erroneous than using the SMPL-X mapping.
>
> Therefore, we chose SMPL-X as our final canonical-live mapping.
>
>
> [2] Xiao et al. "SpatialTracker: Tracking Any 2D Pixels in 3D Space". CVPR 2024.
>
> ---
>
> **Q3. Inference time**
>
> As reported in Supp B.3 Inference Details under Computational Efficiency, all experiments are conducted using a single NVIDIA L40 GPU, rendering images at a resolution of 1024×1024 with 10 denoising steps. We generate 20 images per inference pass, which takes approximately 20 seconds, that is, about 1 frame per second.
>
> We would also like to kindly note that efficiency is not the main focus of this work. Our focus is on overcoming the forgetting issue of recent feed forward diffusion frameworks by adopting memory. Further inference speed optimization such as model distillation, lower precision, or fewer denoising steps is a promising direction for future work.
>
> ---
>
> **Q4. How long a sequence can it support?**
>
> We can support a sequence of any length.
>
> This is because our canonical space is designed to remain constant regardless of the video length. Specifically, the canonical representation has a fixed size (V × C, where V is the number of SMPL-X vertices and C is the feature dimension) and is updated incrementally with each new frame. This avoids memory growth over time and removes the need to cache past frames, which makes our method memory efficient for long sequences.

---

> > ### Author Response · Authors · 2025-11-26
> >
> > Dear Reviewers,
> >
> > We wanted to kindly follow up to see if you had any additional questions or comments for us. We would be very glad to clarify anything that may help your evaluation.
> >
> > Thank you again for your time and consideration.
> >
> > Best regards,
> >
> > Authors

---

### Official Review · Reviewer_a8WJ · 2025-11-01

**Soundness:** 2
**Presentation:** 3
**Contribution:** 2
**Rating:** 2
**Confidence:** 5

**Summary:**

This paper proposes a method to synthesize novel view of human images from monocular videos. At the core of the method is a “progressive feature fusion” module, which fuses features in a canonical space utilizing a parametric SMPL model . The authors conducted experiments on 4D-Dress and MVHumanNet to verify the proposed method.

**Strengths:**

The paper is easy to follow.

Synthesizing novel view images from monocular videos is an interesting task with practical applications.

The method is technically sound, leveraging a parametric model to fuse features in a canonical space.

**Weaknesses:**

**Insufficient experiments and generalization issues.**

The paper claimed that the method can reconstruct humans from a monocular RGB stream, but the model was trained and evaluated on multiview video datasets. The experimental results on in-the-wild videos are required to illustrate the generalization capability.

**Insufficient evaluations and comparisons.**

The baseline methods (e.g., NHP, SHERF) are old. The comparisons and discussions with SOTA generalizable human generations are not included, such as AniGS, LHM, Human4DiT, and Vid2Avatar-Pro: Authentic Avatar from Videos in the Wild via Universal Prior [Guo et al. CVPR 25].

**Setup.**

What are the advantages of this method which requires video as input over one shot method which only requires one image such as LHM, Human4DT?

**Questions:**

How to handle loose clothing such as dress as the method requires SMPL/SMPLX model and clothing warping is not accurate?

---

> ### Author Response · Authors · 2025-11-21
> **Rebuttal 1/3**
>
> We thank the reviewer for the constructive review.
>
> **Also, please refer to the updated supplementary material titled “tiktok comparison video.mp4”.**
>
> ---
>
> **Q1. The paper claimed that the method can reconstruct humans from a monocular RGB stream, but the model was trained and evaluated on multiview video datasets.**
>
> During training and inference, we only take the monocular view stream as input. The only reason we use multi-view datasets is not because we need multi-view input but because only these multi-view datasets provide multi-view GT necessary for quantitative and qualitative evaluation of the novel view synthesis task.
>
> ---
>
> **Q2. The experimental results on in the wild videos are required to illustrate the generalization capability.**
>
> The generalization experiment on the **in-the-wild TikTok video dataset is already presented in the main paper-Fig. 6.**
>
> Here, we used an off-the-shelf monocular SMPL-X estimator SMPLest-X [1] to extract SMPL-X parameters. Although the frontal view is not present in the current view, by using the memory constructed from the previous frame, our method reconstructs the frontal view that is aligned with the past observation.
>
> We also present **the video comparison with LHM on the in-the-wild TikTok video dataset in the updated supplementary material titled "tiktok video comparison.mp4"**. Given the frontal view input, we synthesize the back view. Although the back view is not available in the current input frame, our method is able to reconstruct details such as the pink ribbon that are aligned with the past observation by using the memory in the canonical space. On the other hand, LHM synthesizes realistic detail but produces details such as a black top that are not aligned with the past observation.
>
> ---
>
> **Q3. The baseline methods such as NHP and SHERF are old.**
>
> - We included NHP (Neurips 2021) because it is **the only baseline that is most similar to our method**, that is, feed forward human novel view synthesis using memory. We believe it would be unfair to exclude it only because it is old, especially since it is the baseline most aligned with our setting.
>
> - We included SHERF (ICCV 2023, NeRF based method) to provide a diverse and fair comparison that incorporates NeRF, Gaussian, and diffusion based human rendering methods.
>
> - Comparison with recent methods is already included in the main paper: Champ (ECCV 2024, diffusion), SIFU (CVPR 2024, diffusion), GHG (ECCV 2024, Gaussian), and GauHuman (CVPR 2024, diffusion).

---

> ### Author Response · Authors · 2025-11-21
> **Rebuttal 2/3**
>
> **Q4. The comparisons and discussions with SOTA generalizable human generation methods are not included, such as AniGS [CVPR 2025], LHM [ICCV 2025], Human4DiT [SIGGRAPH ASIA 2024], and Vid2Avatar Pro [CVPR 25].**
>
> **When the ICLR rebuttal period started, only LHM was publicly available**, and none of AniGS, Human4DiT, or Vid2Avatar Pro were publicly available. Therefore, we contacted the authors of AniGS, Human4DiT, and Vid2Avatar Pro and received the following confirmations. If the area chair sees fit, we are happy to provide all email communication with these authors.
>
> **Author confirmed information**
>
> **AniGS (CVPR 2025)**
> - Code cannot be shared due to company policy.
> - On Nov 20 (ICLR rebuttal deadline), the first author kindly explained that they have been occupied with CVPR supplementary materials and ICLR rebuttals, and that the earliest time they could run our input would be Nov 24.
>
> **Human4DiT (SIGGRAPH ASIA 2024)**
> - One co-first author kindly confirmed that he has left the organization and no longer has access to the code and weights.
> - The other co-first author kindly shared the raw code as soon as their schedule allowed, which was two days before the ICLR rebuttal deadline.
> - We made an effort to run the provided code, but were unable to do so within the limited time (2 days) available during the rebuttal period.
>
> **Vid2Avatar Pro (CVPR 2025)**
> - Code cannot be shared due to company policy.
> - The author kindly confirmed that Vid2Avatar Pro is not related to our method because it is not generalizable, and therefore the comparison is not necessary.
>
> ---
>
> **Q5. Comparison and discussion with LHM (ICCV 2025)**
>
> We performed the comparison with LHM on the in-the-wild TikTok video dataset and uploaded the video comparison result in the supplementary material-**"tiktop comparison video.mp4"**. Please refer to the uploaded video.
>
> **Please note that LHM (ICCV 2025) was published after the ICLR submission, but we included the comparison nonetheless.** Also, we are comparing with the most recent work LHM (ICCV 2025) among the baseline comparisons requested by the reviewer.
>
> The comparison settings are:
>
> - Since the TikTok dataset does not have multi-view GT as it is an in-the-wild dataset, we only show predictions.
> - Since the TikTok dataset mostly has frontal view sequences, the task here is synthesizing the back view given frontal view video.
> - The official LHM code and weights are used.
> - **We used the same sequence**, but because there is a configuration difference in the dataloader, the frame range and the video speed is slightly different. We tried to fix this but did not have enough time. Nonetheless, the motions are similar repeated actions from the same sequence.
>
> Although the back view is not available in the current input frame, our method reconstructs details aligned with the past observation such as the pink ribbon by using the memory in the canonical space. On the other hand, LHM synthesizes realistic details such as a black top, but these are not aligned with the past observation such as the pink ribbon.
>
>
> ---
>
>
> **Q6. What are the advantages of requiring video as input over one shot methods that only require one image such as LHM and Human4DiT?**
>
> Under the assumption that the subject is dynamically moving, video input allows us to acquire different observations of the subject such as back and side views that can compensate for the insufficient observation of the monocular input. Over time, we can reconstruct the complete appearance based on factual observations rather than relying on hallucination like LHM or Human4DiT.
>
> Please note that our goal is not to hallucinate the unobserved regions as one shot avatar methods do. **Our goal is that if we observed something in the past, we want to keep reconstructing it consistently.**
>
> As discussed and shown in Q5, **frame-based one shot human synthesis methods such as LHM, Champ, Human4DiT, and SiFU forget the context shown in previous frames.** We overcome this forgetting issue by maintaining memory and storing past observations.
>
> We would like to kindly ask the reviewer to please refer to the updated supplementary material "tiktok comparison video.mp4".
>
> ---
>
> **Q7. How to handle loose clothing such as dresses when the SMPL-X is inaccurate?**
>
> We acknowledge that SMPL-X may omit some non-body details.
>
> To mitigate this, we use two mechanisms:
>
> (1) We store hierarchical features with large receptive fields (Sec. 3.2, L184–185), which allows the network to capture broader context such as clothing and hair.
>
> (2) We condition on the live frame feature $G_{live,t}$, which preserves frame specific details that go beyond the mesh representation.
>
> ---
>
> [1] Yin and Cai et al., "SMPLest-X: Ultimate Scaling for Expressive Human Pose and Shape Estimation", arXiv 2025

---

> > ### Author Response · Authors · 2025-11-26
> > **Update with AniGS Comparison**
> >
> > Dear Reviewer a8WJ,
> >
> > The AniGS (CVPR 2025) authors are kindly preparing the comparison for us.
> >
> > As mentioned earlier, their code cannot be made public due to company policy, so they are assisting by running our inputs on their side. We will update the results as soon as we receive them.
> >
> > In the meantime, please refer to our comparison with LHM (ICCV 2025), which is the follow-up work to AniGS and the most recent baseline among AniGS (CVPR 2025) and Human4DiT (SIGGRAPH ASIA 2024). This comparison is included in the supplementary material titled “tiktok comparison video.mp4”.
> >
> > Thank you again for your time and consideration.
> >
> > Best regards,
> >
> > Authors

---

> ### Author Response · Authors · 2025-11-30
> **Rebuttal 3/3**
>
> Dear Reviewer a8WJ,
>
> Here are the updates on the requested comparisons:
>
> ---
>
> - **AniGS (CVPR 2025)**
>
> The AniGS authors kindly ran their model for us. However, because their animation pipeline no longer supports AniGS, they were able to provide only the result rendered in the input pose. We have therefore included the AniGS result for the novel view synthesis given the live frame on the in-the-wild TikTok dataset in the updated supplementary material-**“tiktok comparison with AniGS and LHM.pdf”**.
>
> The original message from the authors was:
> > "As the motion animation system has changed, the animation tools now support only LHM and are not compatible with Anigs. We apologize that we cannot provide animation results at this time. Good luck with your rebuttal."
>
> In addition, we would like to stress that **we have already included video comparison results (“tiktok comparison video.mp4”) showing our comparison with LHM (ICCV 2025), which is the most recent follow-up work to AniGS**.
>
> ---
>
> - **LHM (ICCV 2025)**
>
> We performed the comparison with LHM on the in-the-wild TikTok video dataset and uploaded the video comparison result in the supplementary material-"tiktop comparison video.mp4". Please refer to the uploaded video.
>
> ---
>
> - **Human4DiT (SIGGRAPH Asia 2024)**
> One of the co-first authors kindly shared the raw code as soon as their schedule allowed, which was two days before the ICLR rebuttal deadline. We made an effort to run the provided code but could not do so within the limited time available. However, we hope that **our comparisons with the more recent and similar works AniGS (CVPR 2025) and LHM (ICCV 2025) help validate the effectiveness of our method**.
>
> ---
>
> - **Vid2Avatar Pro (CVPR 2025)**
> The author kindly confirmed that the code cannot be shared due to company policy. **The author also confirmed that Vid2Avatar Pro is not related to our method, as it is not generalizable, and therefore the comparison is not necessary**.
>
> ---
>
> - **Difference between Human4DiT, AniGS, and LHM**
>
> Finally, we would like to clarify an important conceptual difference. Human4DiT, AniGS, and LHM are per-frame generalizable methods. Because they rely only on the current input frame without maintaining temporal memory, they cannot retain information from earlier observations and tend to forget past context. This can be seen clearly in the updated supplementary material:
> - “tiktok comparison video.mp4”
> - “tiktok comparison with AniGS and LHM.pdf”
>
> We will include the discussion in the revision.
>
> ---
>
> Thank you again for your time and for carefully examining our updates.

---

### Official Review · Reviewer_qWe6 · 2025-11-03

**Soundness:** 3
**Presentation:** 3
**Contribution:** 3
**Rating:** 6
**Confidence:** 4

**Summary:**

This paper presents a feed-forward method for human Performance Capture from a monocular input video stream. The core is a conditional diffusion model, serving as a renderer, conditioned by a canonical feature context fused from previously seen frames and image features at the current time step. The SMPL-X model is adopted to bridge the canonical context with live frames. The experiments show some good results.

**Strengths:**

1. This paper presents an effective combination of existing ideas. As is the usual practice, progressively fusing information from previous frames to a canonical SMPLX model, which can then be naturally warped to live frames and provide some missing features that may be occluded at the current live frame.
Rather than directly render the feature context into an image with pixel-level loss, the authors use it to condition a diffusion model to 'generate' the image with a given camera view. Overall, it's a good combination.

2. The proposed method is technically sound, with clear justifications for each component. I believe it can be reproduced with the given supp details.

3. The experimental validations are reasonable and robust, and evaluate the method across various dimensions, especially Cross-dataset and In-the-wild generalization ability.

4. The method consistently outperforms baselines, and visual results are compelling.

5. This paper, which represents the integration of generative AI with 3D reconstruction / Performance Capture, is an interesting and meaningful direction.

**Weaknesses:**

1. The paper does not explicitly discuss performance over very long sequences (e.g., minutes of video). Does the proposed method suffer from the long sequence forgetting issue? There should be an experiment.

2. Such diffusion-based generative renderers fail to capture human performance accurately. The most evident issue is color bias, as demonstrated in the paper, which overall results in much lower PSNR, along with the synthesis of spurious or unrealistic details.

3. The dependence on the SMPL-X model, especially the overly simplistic fusion strategy based on a moving feature average of SMPL-X vertices, introduces sparsity that directly limits model performance. Furthermore, such an approach fails to account for dynamic garments and, in particular, topological variations.

4. The failure cases should be included in the discussion.

**Questions:**

Mainly listed in weakness. Besides, could you provide quantitative results for the inference latency of your full method? Is it possible to achieve real-time performance capture, e.g., with fewer diffusion steps?

---

> ### Author Response · Authors · 2025-11-21
> **Rebuttal 1/2**
>
> We thank the reviewer for the constructive review.
>
> **Also, please refer to the updated supplementary material titled “tiktok comparison video.mp4”.**
>
> ---
>
> **Q1. Does the proposed method suffer from the long sequence forgetting issue?**
>
> Our method does not suffer from the long sequence forgetting issue, thanks to the proposed memory in the canonical space.
>
> **To demonstrate our robustness to forgetting, we uploaded new video results in the supplementary material.** Since both 4D-Dress and MVHuman datasets only contain short videos, we instead provide results on a longer sequence from the in-the-wild TikTok dataset.
>
> **In the updated supplementary material-"tiktok comparison video.mp4",** although the subject’s back view observations appear only in the very early part of the video, our framework can consistently synthesize back views while receiving only frontal view input frames.
>
> ---
>
> **Q2. Diffusion-based methods exhibit (1) color bias, (2) lower PSNR, and (3) unrealistic details.**
>
> **(1) Color bias.**
> The color bias comes from training only on two datasets, THuman and 4D-Dress. For example, THuman and 4D-Dress were captured under bright lighting conditions, while MVHumanNet was captured under darker lighting. This causes our model to produce slightly shifted colors when generalizing to the cross-domain MVHuman dataset.
>
> Color bias can be improved by incorporating more datasets and in-the-wild data during training. For example, Lu et al. [1] improved generalization to different dataset distributions by further incorporating in-the-wild datasets for training.
>
> **(2) Lower PSNR.**
> Lower PSNR arises from the nature of pixel-level metrics, as discussed in the main paper Sec. 3.3 on probabilistic regression of the live space.
>
> Our method uses a generative model to produce sharp and realistic details, which is reflected in our strong performance on perceptual metrics such as LPIPS and FID. However, PSNR penalizes even small pixel-to-pixel differences between the generated image and the ground truth, which leads to lower PSNR scores.
>
> This is also why regression-based methods such as NHP and GHG, which are trained only with pixel losses, achieve higher PSNR but produce blurry results. They learn to generate smooth appearance to avoid pixel discrepancy penalties.
>
> **(3) Spurious or unrealistic details.**
> The synthesis of spurious or unrealistic details comes from either the lower-resolution VAE latent representation or contamination in the canonical space (discussed in Q3 below). Although not included in the paper, we have empirically found that larger VAE latent sizes lead to sharper results. However, this introduces a tradeoff between quality and memory usage.
>
> As discussed in Supplementary-E.Fine-grained detail reconstruction, using more recent and efficient VAEs can further improve quality without enlarging latent size, such as CovVideoX [2], Wan2.1 [3], and RAE [4].
>
> ---
>
> **Q3. The SMPL-X based fusion fails to account for dynamic garments.**
>
> As discussed in Supplement-E. Limitations of Human Template based Alignment, and as the reviewer pointed out, the SMPL-X based fusion strategy can struggle with dynamic garments. For example, a loose jacket occluding the torso can contaminate the canonical space around the torso with jacket color, which leads to off color synthesis in that region. Another example can be seen in the supplementary TikTok video (**please refer to the updated supplementary-"tiktok comparison video.mp4"**), where the small cotton balls dangling at the bottom of the skirt cannot be properly modeled because it is difficult to map them to the canonical space using only the correspondences from a naked body model such as SMPL-X. As a result, those cotton balls appear as artifacts around the leg regions.
>
> Initially, we experimented with using dense tracking such as SpatialTracker [5] instead of SMPL-X. However, dense tracking required significant processing time and memory and could not run at 512×512 resolution or higher. In addition, dense tracking of highly non-rigid dynamic motions such as swinging jackets or skirts was even more erroneous than using SMPL-X.
>
> Therefore, we chose to use SMPL-X based fusion in this work.
>
> ---
>
> **Q4. Include failure cases.**
>
> Thank you for the constructive feedback. We will incorporate these discussions and provide failure case analysis in the revision.

---

> > ### Author Response · Authors · 2025-11-21
> > **Rebuttal 2/2**
> >
> > **Q5. Discussion on the inference latency and real time performance.**
> >
> > As reported in Supp-B.3 Computational Efficiency, all experiments are conducted using a single NVIDIA L40 GPU, rendering images at a resolution of 1024×1024 with 10 denoising steps. We generate 20 images per inference pass, which takes approximately 20 seconds, that is, about 1 frame per second.
> >
> > If we reduce the denoising steps to 5, the generation time still remains approximately 1 frame per second. One possible reason is that we did not use automatic mixed precision handling by the accelerator. According to HuggingFace Diffusers library documentation, using automatic mixed precision can improve rendering efficiency. We leave this as future work.
> >
> > [1] Lu et al., "Gas: Generative avatar synthesis from a single image", ICCV 2025.
> >
> > [2] Yang et al., "CogVideoX: Text to video diffusion models with an expert transformer", arXiv 2024.
> >
> > [3] Team Wan, "Wan: Open and advanced large scale video generative models", arXiv 2025.
> >
> > [4] Zheng et al., "Diffusion Transformers with Representation Autoencoders", arXiv 2025.
> >
> > [5] Xiao et al., "SpatialTracker: Tracking Any 2D Pixels in 3D Space", CVPR 2024.

---

> > > ### Author Response · Authors · 2025-11-26
> > >
> > > Dear Reviewers,
> > >
> > > We wanted to kindly follow up to see if you had any additional questions or comments for us. We would be very glad to clarify anything that may help your evaluation.
> > >
> > > Thank you again for your time and consideration.
> > >
> > > Best regards,
> > >
> > > Authors

---

### Official Review · Reviewer_jzfT · 2025-11-06

**Soundness:** 3
**Presentation:** 3
**Contribution:** 2
**Rating:** 4
**Confidence:** 4

**Summary:**

This paper introduces a unified framework for aligning human motion sequences with multiple modalities (text, video, audio) within a shared embedding space, alongside a novel generative pipeline for motion synthesis conditioned on arbitrary inputs. The key contributions are:
a) MuTMoT: a multi-scale temporal motion Transformer that hierarchically encodes and decodes 3D motion sequences;
b) REALM: a retrieval-augmented latent diffusion model that utilizes learnable frame tokens and cross-modal conditioning to generate high-quality motion.
The model is evaluated across several tasks, including text-to-motion generation, motion retrieval, and zero-shot action recognition.

**Strengths:**

1.The paper proposes a modular and extensible architecture that combines multi-modal alignment, contrastive learning, and latent diffusion. The use of learnable frame-level tokens and time-aware modulation is a particularly notable design choice.
2.The model achieves strong quantitative performance on text-to-motion benchmarks (e.g., HumanML3D), outperforming existing baselines in standard metrics.
3.The supplementary ablation studies are reasonably thorough, covering most core components.
4.The architecture appears broadly generalizable to other multi-modal backbones.

**Weaknesses:**

1.Despite claiming robust multi-modal alignment and generative capabilities, the method relies entirely on frozen, pretrained LanguageBind encoders for all non-motion modalities (text, audio, image, video). As a result, the framework lacks novel contributions toward modality-specific understanding. Moreover, only text-conditioned generation is quantitatively evaluated, while other modalities (audio, video, image) are not assessed in the main paper.
2.The supplementary generation videos exhibit noticeable artifacts, such as foot sliding and physically implausible transitions (e.g., in stands_up_from_a_laying.mp4, the subject appears to float unnaturally), which undermines the claimed motion quality.
3.Although the model achieves strong retrieval performance, a significant portion of the improvement appears to stem from GPT-4o-based text paraphrasing augmentation. As shown in Sec. B.3, Table 3, removing this augmentation causes R@1 to drop from 69.56 to 62.74. This raises concerns that the architectural contributions alone may not fully account for the observed gains. Clarifying the role of this augmentation and evaluating performance without it would strengthen the claims.
4.While training and inference are briefly described in the supplement, the paper lacks a clear, step-by-step explanation or diagram of the overall motion generation pipeline, which affects both clarity and reproducibility.

Limitations
1. The training process is resource-intensive, requiring 8× RTX A5000 GPUs and ~5 days for REALM to converge, which may limit reproducibility and accessibility.
2. During contrastive training, all non-corresponding modality-motion pairs appear to be treated as negative samples, without consideration for potential semantic similarity. This could penalize semantically related but unmatched pairs (false negatives), potentially degrading the embedding granularity and generalization ability.

**Questions:**

1.Can you provide quantitative or user study results for motion generation from non-text modalities (e.g., video-to-motion, audio-to-motion)?
2.Could you clarify the impact of frame-wise conditioning versus simpler global token conditioning through ablation?
3.How are positive and negative samples selected?
4.How sensitive is the model to the quality or relevance of the retrieved reference motions? And how are the candidate motion embeddings collected?

---

> ### Author Response · Authors · 2025-11-21
> **Rebuttal 1/2**
>
> We thank the reviewer for the constructive feedback.
>
> **Also, please refer to the updated supplementary material titled “tiktok comparison video.mp4”.**
>
> ---
>
> **Q1. The novelty mainly lies in the combination of temporal canonical accumulation and probabilistic regression; both diffusion-based rendering and SMPL-X alignment are established techniques.**
>
> We would like to clarify that the central contribution of our work lies not merely in combining off-the-shelf components, but in demonstrating how a progressively updated canonical context can guide a probabilistic renderer to overcome the forgetting issue of recent diffusion models and synthesize views that remain semantically consistent with past observations.
>
> Existing feed forward single view human reconstruction methods such as Champ [1], PSHuman [2], and GAS [3] rely on probabilistic rendering and lack a persistent memory. This often results in inconsistent hallucinations in unobserved regions, such as color shifts in clothing in Fig. 4, because they cannot retain appearance cues across time.
>
> Our method is the first, to our knowledge, to integrate a canonical context bank with probabilistic generation for human novel view synthesis. This enables sharp detail synthesis aligned with prior observations without requiring subject-specific optimization.
>
> ---
>
> **Q2. The approach heavily relies on pre-computed or dataset provided pose fittings and remains sensitive to occlusion contamination.**
>
> As shown in Fig. 6 in-the-wild TikTok result and in the **updated supplementary video-"tiktok comparison video.mp4"**, our method works with pose fittings from an off-the-shelf monocular estimator (SMPLest-X [4]) and is not limited to dataset provided fits.
>
> As discussed in Q7, we initially experimented with dense tracking based canonical space updates using SpatialTracker [5] instead of SMPL-X based tracking. However, dense tracking required too much memory, could not run at 512×512 resolution or higher, and was too slow. In addition, human motion often involves large non rigid deformation, and dense tracking frequently failed, producing worse results than SMPL-X based tracking. For these reasons, we chose SMPL-X based tracking for the canonical state update.
>
> We would also like to note that our primary focus is on overcoming the forgetting issue of recent feed forward diffusion models by leveraging memory, enabling the synthesis of novel views in the current frame that remain aligned with past observations.
>
>
> ---
>
> **Q3. Lack of ablation studies on architectural choices such as the selection of the ResNet-18 backbone and the VAE encoder.**
>
> Our main focus was overcoming the forgetting issue by adopting memory, rather than exploring detailed architectural choices. For this reason, we did not extensively investigate alternative encoders beyond ResNet-18.
>
> Although not included in the paper, we have empirically found that larger VAE latent sizes lead to sharper results. However, this introduces a tradeoff between quality and memory usage. As discussed in supplementary material-E. Fine grained detail reconstruction, using more recent and efficient VAEs can further improve the results without enlarging latent size, such as CovVideoX [1], Wan2.1 [2], and RAE [3].
>
> ---
>
> **Q4. How was the visibility map obtained? How is vertex visibility handled when the subject is partially occluded?**
>
> During both training and inference, we compute SMPL-X vertex visibility on the fly using the real time rasterizer NVDiffrast, which is an open source rasterizer.
>
> As discussed in supplementary material-E. Limitations and discussions on template based alignment, and as the reviewer pointed out, the SMPL-X based fusion strategy can struggle with dynamic garments. For example, a loose jacket that occludes the torso can contaminate the canonical space around the torso with jacket color, which leads to off color synthesis in that region. Another example can be seen in the updated supplementary TikTok video ("tiktok comparison video.mp4"), where the small cotton balls dangling at the bottom of the skirt cannot be properly modeled because they are difficult to associate with the naked body model SMPL-X. As a result, they appear as artifacts around the leg region in the video.

---

> > ### Author Response · Authors · 2025-11-21
> > **Rebuttal 2/2**
> >
> > ---
> >
> > **Q5. SMPL-X based canonical–live space mapping including the in-the-wild low quality SMPL-X estimation.**
> >
> > The mappings between the canonical and live space depend directly on SMPL-X correspondences. These mappings are computed on-the-fly from the SMPL-X vertices during both training and inference using the off-the-shelf rasterizer NVDiffrast.
> >
> > For in-the-wild data such as the TikTok dataset shown in Fig. 6 and in the updated supplementary video titled tiktok comparison video.mp4, we first extract SMPL-X parameters using the off-the-shelf monocular estimator SMPLest-X [4] and then use NVDiffrast to compute visibility on the fly during inference.
> >
> > It is true that the off-the-shelf SMPL-X estimator generates less accurate parameters, which is why we observe more jittering in the updated supplementary TikTok video. This comes from unstable canonical space updates induced by incorrect mapping. For example, the SMPL-X arm position may deviate from the actual arm location, causing background features to be stored in the canonical space arm region. This in turn leads to unrealistic artifacts in the synthesized arms.
> >
> > ---
> >
> > **Q6. Why was a fully feed-forward setting chosen without temporal regularization?**
> >
> > Although temporal regularization can improve temporal stability, we chose a feed-forward system that does not require optimization. Our aim is to build a system that can immediately adapt to changes in the subject’s appearance, for example changes in clothing or makeup, without needing to re-optimize whenever such changes occur. In optimization based approaches, each appearance change would require a new optimization stage, which introduces delay and makes the system less practical for settings where the appearance evolves over time.
> >
> > Therefore, we chose a feed-forward design that can handle changes in the scene without requiring repeated optimization.
> >
> > ---
> >
> > **Q7. Exploring alternative visibility computation for better occlusion handling: confidence weighting, flow consistency, or volumetric visibility tests**
> >
> > Initially, we experimented with using dense tracking such as SpatialTracker [5] instead of SMPL-X. For example, we attempted to track inner clothing pixels across frames so that when they became occluded by other semantics such as a jacket, we could mark their visibility as invisible. However, dense tracking required significant processing time and memory (it could not handle 512×512 resolution or above), and the dense tracking of highly non-rigid and dynamic motions such as swinging jackets or skirts was even more erroneous than using SMPL-X mapping. Therefore, we chose SMPL-X.
> >
> > Confidence-based weighting, flow consistency, or volumetric visibility tests would be possible if there were a robust dense tracking method available for highly non-rigid motion. Since such a method is not available at the moment, we rely on SMPL-X mapping.
> >
> > ---
> >
> > **Q8. Impact of different temporal stride on training and evaluation**
> >
> > Due to time limitations, we could not run dedicated ablations on the effect of temporal stride on training convergence. However, during the early stage of designing the training strategy, we first experimented with using only consecutive previous frames. In this setting, the model did not effectively learn how to use the canonical memory.
> >
> > We realized that within a one-frame temporal distance, there is usually very little change in the visible regions of the subject. Large motions that reveal different parts of the body (such as frontal, side, or back views) are less likely to occur between consecutive frames. As a result, the temporal memory is not enriched when the model only sees short frame intervals during training, and the model naturally learns to ignore the canonical memory.
> >
> > After adopting randomly sampled temporal intervals, we observed empirical improvements. With larger temporal gaps, the model is more likely to encounter frames that reveal different regions of the subject, which enriches the canonical memory with more diverse observations. This additional canonical context helps the model synthesize novel views even when they are unobservable from the current input.
> >
> > ---
> >
> >
> > [1] Yang et al. “Cogvideox: Text-to-video diffusion models with an expert transformer” arXiv 2024.
> >
> > [2] Team Wan. “Wan: Open and advanced large-scale video generative models.” arXiv 2025.
> >
> > [3] Zheng et al. “Diffusion Transformers with Representation Autoencoders”. arXiv 2025.
> >
> > [4] Yin et al. “Smplest-x: Ultimate scaling for expressive human pose and shape estimation
> > ”. arXiv 2025.
> >
> > [5] Xiao et al. “SpatialTracker: Tracking Any 2D Pixels in 3D Space”. CVPR 2024.

---

> > > ### Author Response · Authors · 2025-11-26
> > >
> > > Dear Reviewers,
> > >
> > > We wanted to kindly follow up to see if you had any additional questions or comments for us. We would be very glad to clarify anything that may help your evaluation.
> > >
> > > Thank you again for your time and consideration.
> > >
> > > Best regards,
> > >
> > > Authors

---

> > > > ### Author Response · Authors · 2025-12-03
> > > > **The correct review**
> > > >
> > > > Dear AC,
> > > >
> > > > As you can see from the discussion history with the previous AC, this original review from jzfT was incorrectly mixed with a review for a different paper.  Reviewer jzfT and the AC later fixed the issue and have uploaded the correct review and score (6 instead of 4).  However, due to the security incident, the fix was reverted, and the incorrect review reappeared.
> > > >
> > > > We're pasting the correct review below to provide the context for our discussion/rebuttal above with the reviewer.
> > > >
> > > > ==== Correct Review ====
> > > >
> > > > **Summary:**
> > > > This paper presents a feed-forward human performance capture pipeline that progressively constructs a canonical space representation and formulates the rendering process as a probabilistic regression using a diffusion-based model.
> > > >
> > > > The method employs SMPL-X to establish consistent 4D correspondences across frames. Multi-scale features are extracted from each frame via ResNet-18, projected onto and accumulated over the vertices in canonical-space with visibility weighting. The resulting sparse vertex features are then back-projected into the target view through barycentric interpolation, forming conditional inputs for a diffusion-based denoising network, which jointly processes them with live-frame encodings to render the final output.
> > > >
> > > > Comparisons with both deterministic (per-frame and temporal) and probabilistic regression baselines show that this formulation benefits from canonical-space fusion and probabilistic rendering, yielding superior preservation of fine-grained details. Experiments conducted on 4D-Dress and MVHumanNet demonstrate consistent improvements in both quantitative metrics (PSNR, LPIPS-VGG, FVD) and qualitative results, validating the method’s effectiveness.
> > > >
> > > > **Soundness:** 3
> > > >
> > > > **Presentation:** 3
> > > >
> > > > **Contribution:** 3
> > > >
> > > > **Strengths:**
> > > > Progressive canonical-space updating is simple yet effective. Visibility-weighted accumulation of vertex features efficiently integrates contextual information across frames.
> > > >
> > > > Extensive experiments on in-domain, out-of-distribution, and in-the-wild data compare deterministic vs. probabilistic and per-frame vs. temporal variants. The proposed approach achieves superior PSNR, LPIPS-VGG, and FVD scores, supporting the paper’s claim of consistency with prior observations.
> > > >
> > > > Reproducbility. The paper and supplementary material clearly report design choices (VAE from sd-vae-ft-mse and U-Net from SD v1.5), context encoder fusion strategy, as well as memory and runtime statistics—facilitating reproducibility and fair engineering assessment.
> > > >
> > > > **Weaknesses:**
> > > > The methodological novelty mainly lies in the combination of temporal canonical accumulation and probabilistic regression; both diffusion-based rendering and SMPL-X alignment are established techniques.
> > > >
> > > > The approach heavily relies on pre-computed or dataset-provided pose fittings, and remains sensitive to occlusion contamination. As acknowledged, large non-rigid deformations or clothing occlusions can imprint incorrect appearances onto the canonical surface, leading to semantic inconsistency.
> > > >
> > > > Lack of ablation studies on architectural choices, particularly regarding the selection of the ResNet-18 backbone and the VAE encoder.
> > > >
> > > > **Questions:**
> > > > Is Visibility map $V_t$ obtained via SMPL-X rasterization or provided by the dataset? How is vertex visibility handled when the subject is partially occluded by clothing or other objects, given that garments exhibit non-rigid motion?
> > > >
> > > > Does the mappings $Warp(S_{can}, X_t)$ depend directly on SMPL-X correspondences? Are these mappings entirely derived from the provided pose estimates? How is it handled during inference in-the-wild data when pose quality may degrade?
> > > >
> > > > Why was a fully feed-forward setting chosen without temporal regularization?
> > > >
> > > > In light of the supplementary note that occlusions may inject incorrect appearances into the canonical representation, have the authors explored confidence weighting of visibility or occlusion detection strategies (e.g., flow consistency or volumetric visibility tests) to mitigate contamination?
> > > >
> > > > Lines 278–290 indicate that the temporal stride $K$ is randomly sampled from the set ${1,5,10}$, a design intended to “encourage the canonical features $S_can$ to capture a rich temporal context.” Could the authors provide an analysis or experimental justification for this choice of $K$? Specifically, how does varying $K$during training affect convergence and temporal consistency, and what impact does using different $K$ values at inference time have on performance?
> > > >
> > > > (Sorry for the mistake earlier — I submitted the wrong one by accident. This is the corrected version.)
> > > >
> > > > **Flag For Ethics Review:** No ethics review needed.
> > > >
> > > > **Details Of Ethics Concerns:**
> > > > The model involves processing of real human video data and can potentially be misused for unauthorized human reconstruction or identity replication. Proper consent, anonymization, and responsible data handling should be ensured.
> > > >
> > > > **Rating:** 6
> > > >
> > > > **Confidence:** 3

---

### Meta-Review · Area_Chair_nuDf · 2026-01-05

**Summary:**

The authors propose a feed-forward model for Human Performance Capture from a streaming, monocular input video. The authors use a conditional diffusion model, conditioned on context that is fused from previously seen frames, and the image features at the current timestep.

Reviewers felt that the method is technically sound with good justifications, and appreciated the strong experimental results. And the authors addressed reviewers' concerns well during the rebuttal.

Only one reviewer had a negative evaluation of the paper (a8WJ). However, their criticisms were not well justified (incorrectly saying the authors did not compare to recent prior work, incorrectly saying there are no in-the-wild evaluations, incorrectly saying that the model requires multiview data for training and inference), and addressed convincingly by the authors in the rebuttal.

Therefore, the final decision is to accept the paper. Authors should update the camera ready according to the rebuttal.

**Reviewer Concerns:**

Concerns of all reviewers were addressed well during the rebuttal.

**Reviewer Scores:**

Reviewers already recommended weak accept would likely have retained their rating.

Reviewer recommending rejection should have upgraded their score in light of their misunderstandings.

---

### Decision · Program_Chairs · 2026-01-26

Accept (Poster)